# Coop: Memory is not a Commodity

**Jianhao Zhang**[*]
OneFlow Research
daquexian566@gmail.com

**Shihan Ma**[*]
OneFlow Research
mmasss1205@gmail.com

**Peihong Liu**
OneFlow Research
peihong.l@outlook.com

**Jinhui Yuan**
OneFlow Research
yuanjinhui@oneflow.org

## Abstract

Tensor rematerialization allows the training of deep neural networks (DNNs) under limited memory budgets by checkpointing the models and recomputing the evicted tensors as needed. However, the existing tensor rematerialization techniques overlook the memory system in deep learning frameworks and implicitly assume that free memory blocks at different addresses are identical. Under this flawed assumption, discontiguous tensors are evicted, among which some are not used to allocate the new tensor. This leads to severe memory fragmentation and increases the cost of potential rematerializations. To address this issue, we propose to evict tensors within a sliding window to ensure all evictions are contiguous and are immediately used. Furthermore, we proposed cheap tensor partitioning and recomputable in-place to further reduce the rematerialization cost by optimizing the tensor allocation. We named our method Coop as it is a co-optimization of tensor allocation and tensor rematerialization. We evaluated Coop on eight representative DNNs. The experimental results demonstrate that Coop achieves up to $2\times$ memory saving and hugely reduces compute overhead, search latency, and memory fragmentation compared to the state-of-the-art baselines.

## 1 Introduction

Recent development of deep neural networks (DNNs) shows a continuous rage on increasing the scale of the network structures, which dramatically improves the capability of the neural network [1, 2, 3, 4]. Training such gigantic models with billions of parameters, however, requires huge on-device memory [5]. Tensor rematerialization, also known as activation checkpointing, is one technique that allows the training of large DNNs under a limited GPU memory budget without compromising the model's accuracy. This technique has been widely applied in the training of large language models such as GPT-3 [1, 6] and LLaMa [3].

The key idea of tensor rematerialization is to evict some intermediate tensors during forward propagation and recompute them as needed during backward propagation, essentially trading recomputing time for memory. Tensor rematerialization can be traced back to checkpointing in reverse-mode automatic differentiation [7, 8, 9], and was first applied to DNN training as a global optimization technique for static computation graphs (SCGs) [10, 11, 12, 13]. Chen et al. ([10]) reduced the memory cost of training a neural network with $n$ layers to $\mathcal{O}(\sqrt{n})$ by dividing the network into segments and dropping the intermediate results within each segment. Checkmate [11] further formalized tensor rematerialization as a mixed integer linear program and used numerical solvers to find an optimal solution. Tensor rematerialization methods that support dynamic computational

---

[*]Equal contribution.

graphs (DCGs) emerge with the advancement of frameworks that utilize DCGs (*e.g.* Pytorch [14]). Megatron-LM [15] uses selective activation recomputation to drop activations that take up large memories but are cheap to recompute in Transformer layers. Dynamic tensor rematerialization (DTR) [16] automatically optimizes memory usage during the training of DCGs by repeatedly searching for and evicting an optimal tensor until the next tensor can be successfully allocated.

However, all of the above tensor rematerialization methods have a similar problem, that is, they assume the memory in deep learning (DL) system is a fungible commodity where free memory blocks at different addresses are identical. Under this assumption, evicting tensors with size $x$ is sufficient for allocating a tensor with size $x$, which is incorrect unless all of the evicted tensors are contiguous. Evicting tensors that do not contribute to a contiguous memory will lead to memory fragmentation, subsequently increasing the cost associated with any potential rematerializations. This motivates us to ask the question: is it possible to release a contiguous memory block that is large enough for allocating a new tensor while the recomputing cost is the minimum?

To answer this question, we proposed **Coop**, a gradient checkpointing scheme that **co-op**timizes tensor rematerialization and tensor allocation. To the best of our knowledge, Coop is the only tensor rematerialization scheme that fully bypasses the incorrect assumption of DL memory system. We propose that a straightforward solution to evicting a set of contiguous tensors is to use a sliding window algorithm. The window is moved within the memory pool until an optimal solution is found. Evicting the tensors in the optimal window will free enough memory while the recomputing cost is the minimum. In addition, unlike prior works that do not optimize tensor allocations, we introduce a smart memory allocator to further reduce the cost of evicting tensors in the sliding window. The smart memory allocator (1) groups "cheap" tensors together and (2) enables recomputable in-place to prevent the disruption of contiguous memory blocks. We show that by co-optimizing tensor rematerialization and memory allocation, Coop reduces memory fragmentation and supports training large-scale DNNs under the most limited memory budget with less compute overhead and lower search latency. For example, when training a 2.7B-parameter GPT-3 style model, Coop reduces the memory fragmentation rate and compute overhead by up to $\sim 5\times$, the search latency by $\sim 10\times$, and the minimum memory budget by 25% compared with DTR. The proposed strategy is model-agnostic and can be universally used to checkpoint any dynamic models in real time. In summary, Coop makes the following contributions:

- We argued for the first time that existing tensor rematerialization methods overlook the memory system during optimization and wrongly assume that the memory in DL systems is a fungible commodity. We addressed this problem by co-optimizing tensor rematerialization and tensor allocation, which opens up a new perspective on improving checkpointing techniques in DL frameworks.

- Coop allows the tensor rematerialization scheme and memory allocator to facilitate each other. The heuristic of tensor rematerialization is reduced by taking into account the properties of memory allocators, and the memory allocators are improved by considering the efficiency of different operations in tensor rematerialization.

- We demonstrated that Coop enables the training of the most widely-used DNNs with less compute overhead, lower minimum memory budget, and smaller search latency.

- Coop has been implemented in OneFlow framework and it can be easily integrated into any other deep learning framework.

## 2    Background and Motivation

### 2.1    Memory Allocator in DL Systems

Memory allocator is an important component of deep learning systems. All known DL systems (*e.g.* PyTorch [14], TensorFlow [17], and MXNet [18]) have their own memory allocators to enable fine-grained memory management and avoid the overhead of communication with the operating system (OS). Compared to advanced CPU memory allocators, *e.g.* mimalloc [19] and jemalloc [20], memory allocators in DL frameworks are simpler. Typically, in DL frameworks, when a tensor is destructed, its memory is not returned to the OS but inserted into the *free list* of the allocator and merged with other chunks with adjacent memory addresses. When the user requests memory for a tensor with size $S$, the allocator tries to place the tensor at the leftmost side of a free chunk with a size equal to or larger than $S$. If the sizes of the free chunks are all less than $S$, the allocator will

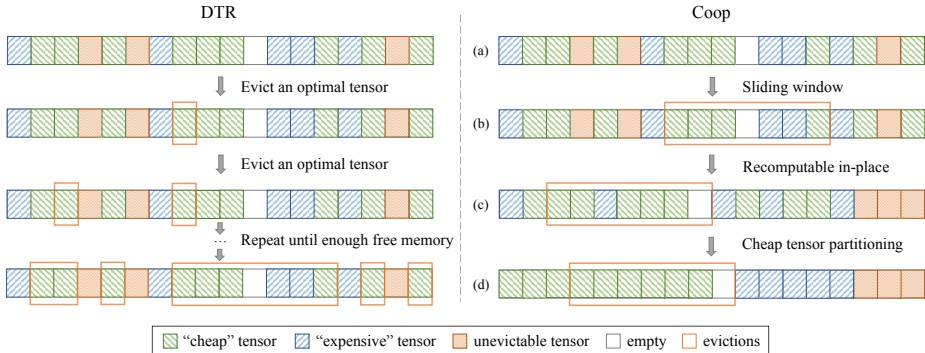

Figure 1: Comparison between DTR and Coop. DTR overlooks the underlying memory system, which leads to redundant evictions. In contrast, Coop co-optimizes tensor rematerialization and tensor allocation to find the best tensors to evict. (a) A typical memory layout given by the conventional tensor allocator, where all tensors are mixed together. Tensors are classified by their *cost densities* (computational cost divided by memory size). The "cheap" and "expensive" tensors denote the tensors with low and high cost densities, respectively. Unevictable tensors, including network parameters and buffers, cannot be evicted. (b) Coop uses the sliding window algorithm to find the optimal set of tensors under a given memory layout. (c) Coop uses recomputable in-place operations to optimize memory layout by ensuring that the unevictable tensors remain in the same place even after being updated. This optimization reduces the cost of the tensors in the sliding window compared to (b). (d) With cheap tensor partitioning, "cheap" and "expensive" tensors are separately allocated to the two sides of the memory pool. The memory layout is further optimized with the eviction cost reduced.

request new memory from the OS, even though the total size of the free chunks might be greater than $S$. These unused free chunks are called *memory fragments*. During training, all intermediate activations and gradients are released at the end of each iteration, which produces large free memory chunks. Therefore, tensors generated by sequential operations tend to be allocated to contiguous spaces from the leftmost of the memory pool in each iteration.

Besides, it is possible for memory allocators to leverage the page table of the hardware and re-compose discontinuous memory chunks into a chunk continuous from the virtual memory's view. Studies have proven that memory fragmentation in CPU can be reduced in this way [21, 22]. However, the driver of NVIDIA GPUs is a proprietary NVIDIA-controlled black box, making it almost impossible for community researchers to apply similar ideas to GPUs.

## 2.2 Tensor Rematerialization and Memory Fragmentation

Tensor rematerialization in DNN training is first introduced by Chen et al. [10]. They divide the network into segments, evict all intermediate features within each segment, and recompute them during backpropagation. Subsequent studies enhance this method by intelligently selecting tensors to evict. Dynamic Tensor Rematerialization (DTR) [16] devises a *heuristic* to guide the eviction of tensors during runtime. The heuristic leverages the metadata of each tensor (staleness, memory, and computational cost) to select the stalest, largest, and cheapest tensor to evict when the memory runs out. This process runs several times until the released memory is sufficient for the new tensor. Recent work MegTaiChi [23] proposed Dynamic Tensor Evicting (DTE) based on DTR. DTE modifies DTR's heuristic by considering the adjacent free memory blocks of each tensor.

These tensor rematerialization methods enable the training of large DNNs with limited memory budgets and low overhead. However, their performance is inhibited by large memory fragmentations. As a real-world example, DELTA [24] implements DTR in Parrots framework [25] and reports that the memory fragmentation hugely affects the maximum capacity of the model. Memory fragmentation is the small and sparsely scattered memory chunks that are unusable for tensor allocation. This problem affects all DNN training but is exaggerated by tensor rematerialization, which incurs a more frequent rearrangement of the memory. A fundamental reason why the existing tensor rematerialization methods cannot address this problem is that they do not emphasize producing a

contiguous chunk of free memory. Additionally, their heuristics render the condition worse. Most rematerialization methods penalize evicting tensors generated by sequential operations to reduce recursive rematerializations. For example, the projected cost of a tensor in DTR's heuristic is calculated over the tensor's neighborhood (tensors that depend on or are relied on recomputing the current tensor). Penalizing evictions of sequential tensors often results in discontiguous free memory chunks, as the sequential tensors tend to locate contiguously (see 2.1). To address the memory fragmentation in tensor rematerialization, Coop proposes to evict a set of contiguous tensors by using a sliding window algorithm, which is hugely different from the modification of heuristic as in DTE.

## 2.3  In-place Mutation in DL Systems

In-place mutation is a common and important feature in deep learning frameworks. For example, PyTorch users often construct their networks with in-place mutating operations like `nn.ReLU(inplace=True)`, `tensor.zero_()`, `torch.add(a, b, out=c)`, to avoid allocating memory for the output tensor by directly overwriting the existing tensor. In-place operations are more cache-friendly and use less memory than their out-of-place counterparts. The in-place operations produce correct results as long as the original values are not used in the future. In deep learning frameworks, this requirement is met by not capturing the original value during backward propagation. However, in frameworks with tensor rematerialization, the original value is likely to be used during recomputation, which leads to incorrect results. For example, if the user runs two operations `y = x + 1` and `x.relu_()` in sequence and `y` needs to be rematerialized afterward, the operation `x + 1` will run again but the original value of `x` has already been discarded by `x.relu_()`.

DTR solves this problem by introducing a copy-on-write layer, which copies the tensor and mutates the copied one. This actually transforms the in-place operations into out-of-place operations, which negates the advantages of in-place mutations and may cause potential problems (explained in Section 3.5). Recently, Koka [26], a functional programming language, proposed a novel programming paradigm called functional but in-place (FBIP) in their new memory management system Perceus [27]. FBIP enables in-place mutation in a purely functional manner by allocating new objects to the memory of known unused objects. Inspired by FBIP, we proposed a recomputable in-place module in Coop to prevent unnecessary evictions and reduce memory fragmentation.

## 2.4  Other Memory-saving Techniques

In addition to tensor rematerialization, several other techniques can be used to reduce the memory requirements for training large models, such as model parallelism [28], gradient accumulation, reversible operations [29, 30], quantization [31], and offloading [6, 32]. These methods are complementary to tensor rematerialization. The combined use of these techniques can result in more significant memory savings [33], as demonstrated in Turing-NLG, GPT-3, and LLaMa [6, 3].

Nonetheless, each of these techniques comes with its own limitations. Model parallelism generates additional communication overhead [34]. Gradient accumulation necessitates a reduction in batch size, which can adversely affect training accuracy, particularly for models with batch normalizations [35, 36]. Reversible operations do not possess universal applicability. Quantization may entail lossy compression [37]. Offloading is typically slower than rematerialization and requires prefetching for satisfactory performance [33], making it challenging to implement in DCGs.

## 3  Methods

### 3.1  Problem Formulation

Coop is a runtime memory manager that allows the training of dynamic DNNs under a preset memory budget by manipulating the state of the memory pool. The state of the memory pool is denoted as *memory layout*, including the position and the size of each tensor. Coop manipulates the memory layout by intercepting tensor allocation, eviction, and rematerialization.

Assume that there are $N$ tensors in the memory pool after several operations in the forward process. Each tensor $t$ has memory size $m(t)$ and heuristic $h(t)$. Heuristic is the cost of evicting a tensor (defined in 3.3), the less the better. Coop aims to find an optimal set of tensors to evict when the memory meets the budget, such that (1) evicting the set of tensors is sufficient for allocating the new

tensor and (2) the evictions bring in the minimum potential rematerialization cost. Guided by this target, we introduce the cost function of Coop as follows:

$$\underset{S,L}{\arg\min} \sum_{t \in S} h(t), \text{ subject to } M(S,L) \geq M_R \tag{1}$$

where $L$ denotes a memory layout, $S$ denotes a set of tensors, $M(S,L)$ is the largest available contiguous memory under memory layout $L$ after evicting all tensors in $S$, and $M_R$ is the required memory size.

## 3.2 Method Overview

We observed two desiderata from the cost function, which inspired us to propose the three modules in Coop. First, free but not contiguous memory blocks in $S$ do not contribute to the largest contiguous memory $M(S,L)$ and thus will not be used for allocating new tensors. In other words, it is more efficient to evict tensors that produce a single contiguous block. We used a **sliding window algorithm** to address this problem in $O(N)$ time complexity (3.3). Second, memory layout $L$ affects the cost of evicting tensors in $S$, and $L$ is related to tensor allocation in DCGs. Given that a contiguous block can be produced by the sliding window algorithm, we further introduced two modules to reduce the cost of evicting tensors in this block by managing the memory layout: (1) we proposed **cheap tensor partitioning** in 3.4 to cluster tensors that are computationally "cheap" to the same place. (2) we introduced **recomputable in-place** in 3.5 to avoid the random reordering of the memory layout that splits contiguous memory. An overview of Coop is displayed in Figure 1, and the algorithm of Coop is shown in Algorithm 1.

---

**Algorithm 1** The algorithm of Coop

**procedure** ALLOCATE($op, size$)
    **if** $op$ is an in-place operation **then**
        $addr \leftarrow input.addr$
    **else**
        $block \leftarrow find\_free\_block\_not\_less\_than(size)$
        **if** $block$ is None **then**
            $evict(sliding\_window\_search(size))$
            $block \leftarrow find\_free\_block\_not\_less\_than(size)$
            **if** $op$ is expensive **then**
                $addr \leftarrow block.left\_addr$
            **else**
                $addr \leftarrow block.right\_addr - size$
            **end if**
        **end if**
    **end if**
    **return** $addr$
**end procedure**

---

## 3.3 Sliding Window Algorithm

A brute-force approach to solving Equation 1 is to traverse the combinations of all tensors and find a valid combination with the minimum cost. It requires a total of $2^N$ enumerations ($N$ is the number of tensors in the memory pool), which is computationally unfeasible. To circumvent this issue, we used the sliding window algorithm to find the optimal solution and reduced the time complexity from $O(2^N)$ to $O(N)$.

We track the states of all tensors by storing them in a list, sorted by the memory addresses. The free memory chunks are included as "special tensors" of which the heuristics are zero (no cost to recompute them). In this way, finding a set of tensors that produce a single contiguous block is equal to finding a subsequence of the list. We use two pointers to denote the start and the end of the sliding window. Tensors in the window are supposed to be evicted. By moving the window within the list and continuously comparing the summed heuristics of the tensors in the window, a set of contiguous tensors of which the summed memory is larger than required and the cost minimum can be found.

To calculate the eviction cost, we need to choose a heuristic that is consistent with our sliding window algorithm. We define the heuristic as $h(t) = \frac{c(t)}{s(t)}$, where $c(t)$ is the projected cost (the sum of the computational cost of tensor $t$ and $t$'s evicted neighborhood) and $s(t)$ is the staleness. Different from DTR's heuristic ($h(t) = \frac{c(t)}{m(t) \cdot s(t)}$, where m(t) is the memory size of $t$), our heuristic does not need the information of tensor's memory size. The reason is that the information of tensor memory has been embedded into the layout $S$, thus the list maintained by Coop. The sliding window algorithm utilizes this information and searches for an exact set of contiguous tensors to evict rather than repeatedly and blindly looking for tensors that might be large enough to cover the new tensor.

One benefit of the proposed sliding window algorithm is that traversing for one time gives the best tensors to evict. Therefore, the time complexity for searching is $O(N)$. Another particular advantage is that no tensors are "innocently" evicted, as the freed memory after each eviction is contiguous and meets the required memory exactly in the value. By contrast, DTR evicts one tensor each time, resulting in sparsely distributed free memory blocks that cause the memory fragmentation issue.

### 3.4 Cheap Tensor Partitioning

Given that a contiguous free memory chunk can be produced by the proposed sliding window algorithm and the overall heuristic is the minimum under the current memory layout, we ask whether it is possible to optimize the memory layout as well to further reduce the overall heuristic. We propose an approach called cheap tensor partitioning to address this problem. Instead of allocating tensors contiguously as in the traditional DL systems, we group tensors with the same magnitude of eviction cost to the same place. Clustering tensors with the same magnitude reduces the overall heuristic in two ways: (1) it breaks the pattern that tensors generated by sequential operations are also allocated contiguously in memory, thereby reducing the projected cost $c(t)$ in the heuristic; (2) it increases the chance to evict a set of contiguous "cheap" tensors. In this way, a proper memory layout is prepared before tensor eviction.

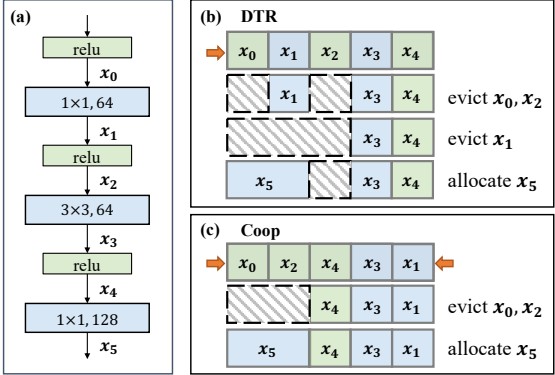

Figure 2: **Illustration of cheap tensor partitioning in Coop.** (a) Layers of a representative convolutional neural network, with convolutions followed by activation layers. (b) Tensor eviction under DTR with tensors allocated in sequence. (c) Tensor eviction under Coop with cheap tensor partitioning. Tensors are allocated from both sides of the memory pool.

We implemented cheap tensor partitioning by allocating tensors from the leftmost and rightmost of the memory pool (Figure 2(c)), which is feasible because the memory budget is given by the user before training. Cost density (computational cost divided by memory size) is used to evaluate the magnitude of eviction cost. Tensors with the same magnitude of cost density are allocated to the same end of memory pool during the forward propagation of the model. We observed that most of the operators in NN can be simply classified into two categories by their complexity: super-linear (*e.g.* matmul and conv, denoted as $\mathcal{C}_1$) and linear/sub-linear (*e.g.* element-wise ops, denoted as $\mathcal{C}_2$). For most cases, the cost density of super-linear operations is larger than the other operators by one order of magnitude. Cost densities of major operators in DNNs on NVIDIA GeForce RTX 2080 Ti are summarized in Table 1.

Figure 2(a) shows an example of a typical convolutional neural network, in which each convolution layer is followed by an activation function. Here we did not include Batch Normalization as it has a

Table 1: Cost density of major operations in DNNs ($\mu s$/MB).

| Operation | ResNet-50[*] | GPT-2[†] | U-Net[*] | Swin-T[†] |
|---|---|---|---|---|
| $\mathcal{C}_1$ Conv/MatMul | 35.6 | 33.5 | 89.3 | 32.7 |
| $\mathcal{C}_2$ Batch/LayerNorm | 5.0 | 4.2 | 5.3 | 4.1 |
| $\mathcal{C}_2$ ReLU/GELU | 3.9 | 3.8 | 3.9 | 3.9 |

[*] Model with Conv, BatchNorm and ReLU
[†] Model with MatMul, LayerNorm and GELU.

similar computational cost density with the activation layers (Table 1), and is not necessarily included in all neural networks. Suppose the tensors $x_0, ..., x_4$ are generated at the start of training, thus stored sequentially in memory. If the memory is full and another 100 MB for $x_5$ is required, the two tensors generated by the two activation layers ($x_0$ and $x_2$, each 50 MB) will be first evicted under DTR's strategy ($x_0$ is the stalest and cheapest tensor in the memory and its eviction increases the heuristic of $x_1$). However, as the freed chunks are not contiguous, extra tensors are required to be freed, *e.g.* $x_1$, bringing in useless evictions and leading to memory fragmentation (Figure 2(b)).

## 3.5 Recomputable In-place

When dealing with operations that directly mutate the content of the input tensor, *i.e.*, in-place operations, DTR introduces a copy-on-write layer to replace the in-place operation with three separate operations: copying the input tensor, mutating the copied tensor, and evicting the input tensor (Figure 3 (c)). Allocating new memory for the copied tensor has two detrimental effects: (1) if the memory pool is full, additional tensors are required to be evicted for allocating the newly copied tensor, which takes time and brings extra recomputation cost; (2) more critically, if this tensor is unevictable, such as the parameter that is updated in-place after each iteration, a random allocation of this tensor will partition the memory into non-contiguous parts, preventing the creation of contiguous free memory.

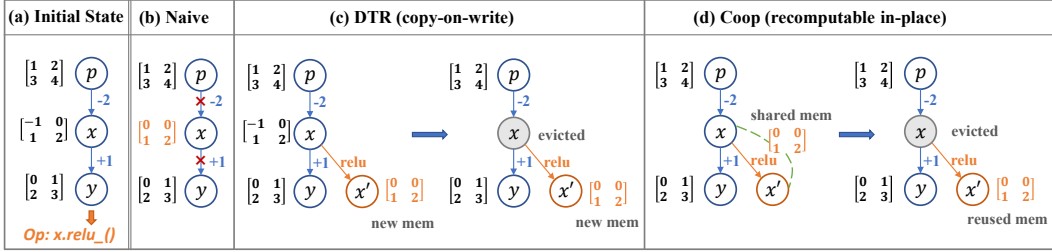

Figure 3: **Illustration of different in-place mechanisms on `relu_` (in-place ReLU).** (a) Initial state of the operations. (b) Unrecomputable naive in-place operations. The dependencies between adjacent variables are incorrect. (c) Copy-on-write mutation in DTR. (d) Recomputable in-place in Coop that avoids allocation of new tensors.

Inspired by the functional but in-place paradigm in Perceus [27], Coop proposes a memory-reuse mechanism named recomputable in-place to handle in-place operations and further optimize the memory layout. Instead of copying the input tensor to a new one, we reuse the memory of the input tensor and directly allocate the new tensor to its original place such that the input tensor and the new output tensor share the same memory (Figure 3 (d)). As the input tensor is evicted after the in-place operation, recomputing tensors that rely on it requires recomputing the input tensor itself first. Therefore, the value of the input tensor is always the same as its initial version.

Besides, we observed that the in-place operations mostly happen during the updates of model parameters that are unevictable. To prevent these tensors from partitioning the whole memory pool into discontinuous chunks, Coop first allocates the tensors of parameters to the two ends of the memory pool, and then reuses their memory during the updates (Figure 1(c)). In this way, the rate of memory fragmentation can be reduced to the least.

# 4 Evaluation

## 4.1 Evaluation Setup

We compared our proposed method, Coop, with two state-of-the-art DCG-based automatic rematerialization methods, DTR [16] and DTE (the rematerialization method introduced in MegTaichi [23]), across eight representative DNNs. The eight DNNs include a 2.7B parameter GPT-3 style large language model, Swin-Transformer, ResNet-50, etc. Additionally, we compared our method with selective activation recomputation [38] (the rematerialization method designed for Transformer models in Megatron-LM[28]) on Transformer models. Comparisons with static graph methods are beyond the scope of this paper due to the distinct prerequisites and use cases.

Table 2: Rematerialization methods compared in evaluation.

| Method | Description | Automatic | MS*-aware | MS-optimized | Traversal Count |
|--------|-------------|-----------|-----------|--------------|-----------------|
| **Coop** | The proposed tensor rematerialization method. | ✓ | ✓ | ✓ | Single |
| DTE | Our impl. of Dynamic Tensor Evicting [23] in OneFlow. | ✓ | ∼⋆ | ✗ | Multiple |
| DTR | Our impl. of Dynamic Tensor Rematerialization [16] in OneFlow. | ✓ | ✗ | ✗ | Multiple |
| SAR | Our impl. of Selective Activation Recomputation [38] in OneFlow. | ✗ | ✗ | ✗ | - |

  * MS is the abbreviation of the memory system.
  ⋆ DTE's heuristic considers adjacent free memory but cannot promise a contiguous memory block is obtained.

Table 2 provides an overview of the methods compared in our evaluation. The baseline methods (DTR, DTE, and SAR) have been implemented in different frameworks. For a fair comparison, we re-implemented all baselines in OneFlow [39], which is an open-source deep learning framework with PyTorch-aligned APIs. In our DTE implementation, we only consider the cost of recomputation, whereas the original paper also considers the cost of swapping tensors between GPUs and the host [23]. We set the coefficient of *computing times* in DTE to 1 to align with DTR and Coop. We also compared Coop in OneFlow with the official DTR in PyTorch and the official DTE in MegEngine in Appendix A. The effects of the three proposed modules were examined in Appendix B. All experiments were conducted on a machine equipped with 4 NVIDIA A100 GPU (80 GB, CUDA 11.7, CuDNN 8.5.0) and 56 Intel(R) Xeon(R) Platinum 8336C CPU cores running Ubuntu 20.04. For BERT Large and GPT-3 style 2.7B, Adam optimizer was used, while the SGD optimizer was used for the other experiments. ZeRO stage 2 [40] is used when training the GPT-3 style 2.7B model. Among the eight DNNs, BiLSTM and SPOS have dynamic network structures that vary based on the input.

## 4.2 Evaluation Metric

Five metrics are used for evaluating the performance of the methods in Table 2, including compute overhead, minimum memory budget, search latency, memory fragmentation rate, and cutoff memory budget. For all of the five metrics, lower is better. Compute overhead is the additional computation time due to tensor recomputation. Minimum memory budget is the minimum memory that the neural network can be trained on. Note that it does not include the memory of CUDA context. Search latency is the time to search for the optimal tensors to evict until the freed memory is sufficient for allocating the next tensor. Memory fragmentation rate is defined as the size of free memory divided by the memory held by the memory allocator. Cutoff memory budget is the memory threshold, below which at least one tensor is evicted during the training process. The experimental results of the cutoff memory budget are displayed in Appendix C.

## 4.3 Results and Discussion

**Compute Overhead and Memory Budget.** Figure 4 displays the compute overhead across various memory budgets and the lowest memory budget for all methods. Coop presents the lowest compute overhead and can be trained under the minimum memory budgets on all eight networks. This improvement is substantial in most networks. For instance, the overhead of training BERT Large

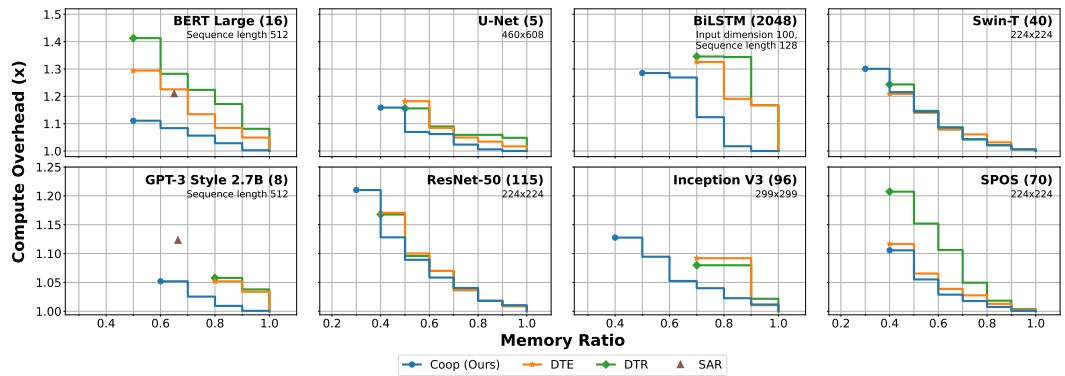

Figure 4: Comparison among different tensor rematerialization strategies on eight DNNs, showing the ratio of compute overhead under different memory budgets (fractions of the original peak memory usage). Model batch sizes are given in parentheses. SAR (selective activation recomputation) is only available on transformer models (BERT Large and GPT-3).

under a 50% memory ratio with Coop is 11%, compared with 29% with DTE and 41% with DTR. Additionally, Coop can save up to 60% of memory usage when training Inception V3, while both DTR and DTE only save half of that amount.

We observed that the runtime overhead of training ResNet-50, Inception V3, and Swin-T is low for all three strategies under high memory ratios. However, the overheads of training the other DNNs, such as U-Net, BiLSTM, and BERT Large, are substantially higher with DTR and DTE than with Coop even at a 90% memory ratio. The reason is that the memory size of the intermediate feature is decreasing in ResNet-50, Inception V3, and Swin-T, which is typical for image classification tasks. Evicting an old single tensor with a large size is often sufficient for allocating a new tensor with a small size. By contrast, BERT Large and BiLSTM have features with consistent sizes and U-Net has a unique U-shaped architecture. In these cases, DTR and DTE need to run in loops to release enough memory, which hugely increases the runtime overhead.

**Search Latency.** As shown in Figure 5, the search latency of Coop is less than DTR and DTE in most cases. Moreover, the search latency of DTR and DTE displays considerable variations across the eight DNNs. The maximum search latency for training BiLSTM exceeds $10^4 \mu$s using DTR, which is five orders of magnitude slower than the 0.32 $\mu$s for training ResNet-50. In contrast, Coop yields more consistent search latency across different DNNs. This is because Coop only requires a single search using the efficient sliding window algorithm, and thus the latency remains low and is not affected by the fluctuations in the size of the intermediate tensors.

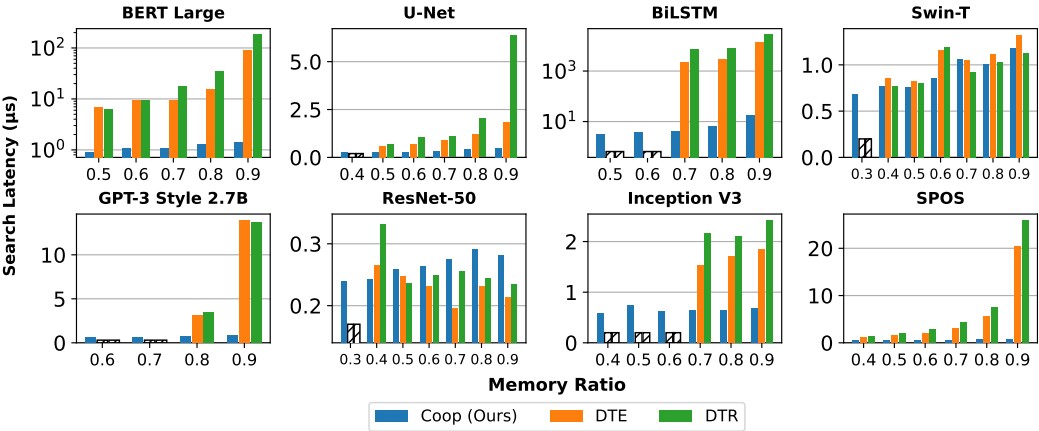

Figure 5: Search latency of Coop, DTR, and DTE under different memory ratios and networks. The bars with slashes represent the OOM error. The y-axes of GPT-2, BERT Large, and BiLSTM are in a logarithmic scale.

We observed that Coop's search latency is larger than DTR and DTE on ResNet-50 under high memory ratios. We surmise this is the result of different engineering implementations rather than the strategies. During the training of ResNet-50, evicting a single resident tensor is sufficient for allocating a new one. Therefore, the search latencies of ResNet-50 are already very low (less than 0.4 $\mu$s), and thus the influence of the implementation is more obvious.

**Memory Fragmentation.** As shown in Figure 6, training the eight DNNs using DTR leads to considerable memory fragmentation rates. Although DTE mitigates memory fragmentation compared to DTR in most cases, its highest memory fragmentation rate remains considerably high. In contrast, Coop significantly reduces the memory fragmentation rate to less than 5% across all DNNs and memory budgets. The performance of Coop is substantially better than DTR and DTE, particularly for BiLSTM and BERT Large models. This demonstrates that Coop utilizes the memory resources more efficiently and explains why the same DNNs can be trained with less compute overhead and lower memory budgets with Coop (Figure 4).

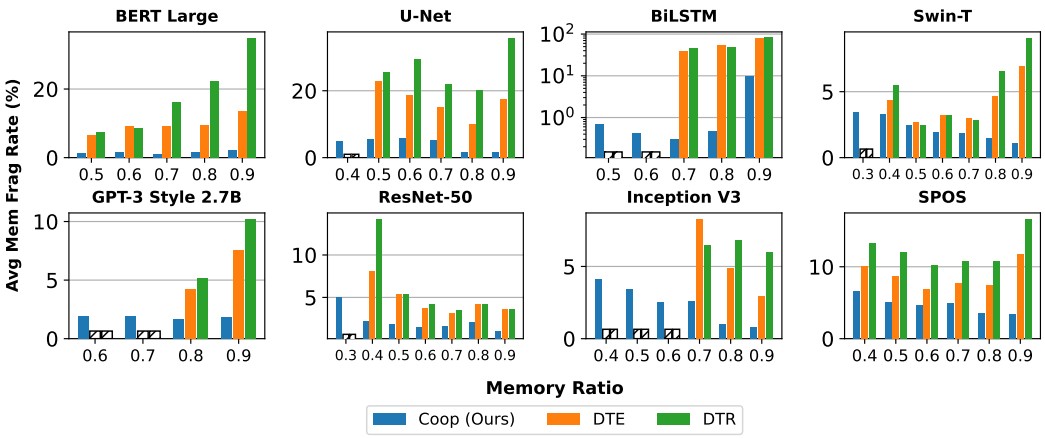

Figure 6: Averaged memory fragmentation rate under Coop, DTR, and DTE. A lower rate denotes more efficient utilization of the runtime memory. The bars with slashes represent the OOM error.

# 5 Limitation

Coop comes with its own memory pool, so it cannot be simultaneously used with CUDA's built-in memory pool (stream-ordered memory allocator). The advantage of using the stream-ordered memory allocator is that multiple programs that use the stream-ordered memory allocator can share the same memory pool. However, all existing deep learning frameworks use their own memory pools instead of CUDA's built-in memory pool in default to achieve better efficiency and flexibility.

Additionally, Coop is an online method so it inherits the pros and cons of online methods. It can provide an efficient solution to finding the best tensors to evict within negligible time. However, the solutions may not match the optimal solutions as solved by the offline methods such as Checkmate, even though these offline methods usually require additional solvers and take several hours or multiple days.

# 6 Conclusion

We proposed Coop to address the challenge of training large dynamic DNNs under restricted memory budgets. We argued for the first time that the memory system in DL frameworks should not be ignored in tensor rematerialization and the conventional checkpointing algorithms on DCGs can be hugely improved by co-optimizing tensor allocation and tensor rematerialization. Leveraging the contributions of the sliding window algorithm, cheap tensor partitioning, and recomputable in-place, Coop demonstrates consistent and superior performance over the state-of-the-art checkpointing algorithms on most of the commonly used models, with the lowest memory budgets, least runtime overhead, and minimum memory fragmentation rate.

## Acknowledgement

The authors thank Marisa Kirisame for discussing with us the idea and paper-writing of Coop, and thank Kan Wu, Qiaoling Chen, and Yipeng Li for their helpful advice. Jianhao Zhang would like to express heartfelt appreciation to his wife, Xingzi Yao, for her unwavering support and companionship. This work was supported by the Major Scientific Research Project of Zhejiang Lab (No.2019KD0AD01) and National Natural Science Foundation of China under Grant No. U20A20226.

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
