# A    Comparison with official DTR and DTE implementations

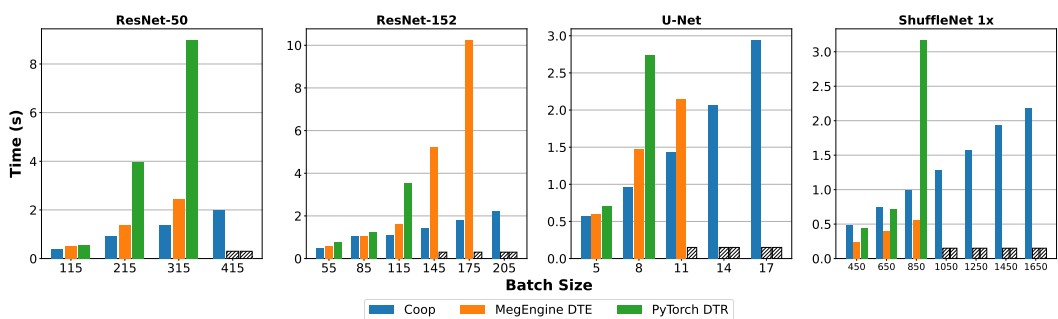

Figure 1: Training time with increase of batch size under Coop (implemented in OneFlow), MegEngine DTE (official implementation of DTE in MegEngine) and PyTorch DTR (official implementation of DTR in PyTorch). The bars with slashes represent the out of memory (OOM) error.

We compared the performance of Coop with the official implementation of DTE in MegEngine (MegEngine DTE) and DTR in PyTorch (PyTorch DTR) on ResNet-50, ResNet-152, U-Net, and ShuffleNet 1x with an NVIDIA RTX 2080Ti GPU. The reason for choosing these four DNNs is that it is difficult to manually implement and train the models that are not available in MegEngine, due to MegEngine's underdeveloped ecosystem. OneFlow, MegEngine (version 1.11.0), and PyTorch (https://github.com/pytorch/pytorch/pull/42056) were compiled with CUDA 11.7 and CuDNN 8.5.0. We modified the source code of PyTorch to allow the training of ShuffleNet with depthwise convolution. The officially released DTE in MegEngine does not include the cost of swapping in the heuristic ([1]) and does not support the settings of memory budget. We have confirmed with the authors of MegEngine that GPU memory can only be fully utilized during training, and thus we did not set the memory budget during the experiments but studied the averaged time of training one iteration while increasing the batch size.

Figure 1 shows the comparison of training time of Coop, DTR, and DTE for four deep neural networks. Coop is significantly faster than the other two strategies during the training of ResNet-50, ResNet-152, and U-Net with different batch sizes. Even though training ShuffleNet 1x with Coop takes more time than with MegEngine DTE, Coop supports the largest batch size, indicating that Coop saves more memory and supports the training of larger models.

Notice that the results in Figure 1 only provide limited information as Coop, DTR, and DTE were separately implemented in three different frameworks where the operators are optimized differently. To give an example, when ShuffleNet 1x was trained with batch size 450, no rematerialization was triggered with Coop, MegEngine DTE, and PyTorch DTR. However, training with MegEngine DTE was one time faster than with the other two strategies. One possible reason is that MegEngine provides highly-optimized CUDA kernel for running depthwise convolution in ShuffleNet.

## B    Ablation Studies

Coop is composed of three modules: (1) sliding window algorithm, (2) recomputable in-place, and (3) cheap tensor partitioning. To analyze the contributions of these three modules, we compared the compute overhead and memory fragmentation rate under Coop with one of the three modules individually removed at each time. When the sliding window algorithm is removed, we replaced the heuristic of Coop with a baseline heuristic, the heuristic of DTE (with free neighbors taken into account).

**Compute Overhead** Figure 2 shows that running Coop with any one of the three core modules removed degrades the performance of checkpointing, resulting in a larger runtime overhead and a higher lowest memory budget that Coop supports. For example, ResNet-50 can be trained under 30% memory ratio with Coop, while the lowest budget is increased to 40% when any one of the modules is removed. Excluding the sliding window algorithm even doubles the overhead of training U-Net under 40% memory ratio. We find that the sliding window algorithm affects the compute overhead

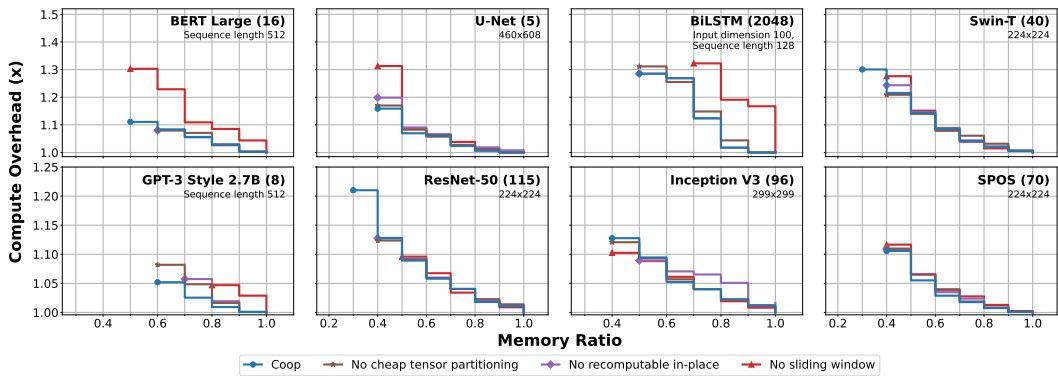

Figure 2: Comparison of compute overhead evaluated on Coop when one of the three modules is removed.

the most, followed by recomputable in-place and cheap tensor partitioning. Cheap tensor partitioning has comparatively less influence, as this module only contributes to the optimization of the memory layout at the beginning of the forward process before the memory pool is full. The lower the memory ratio, the less time during which the cheap tensor partitioning works.

**Memory Fragmentation** Figure 3 shows the impact of the three modules on averaged memory fragmentation rate. Removing either recomputable in-place or sliding window algorithm increases the fragmentation rates. The reason is that both of these two modules avoid useless evictions that bring in memory fragments. The impact of recomputable in-place is less, as this module only affects the in-place operations. Cheap tensor partitioning optimizes tensor allocations but not changes the number of free memory chunks in the memory pool. Even though an optimal tensor layout brought by cheap tensor partitioning could reduce the heuristic of an optimal eviction, the fragments due to the mismatch between the freed memory and the required memory might not be reduced. As a result, cheap tensor partitioning does not directly affect the memory fragmentation rate.

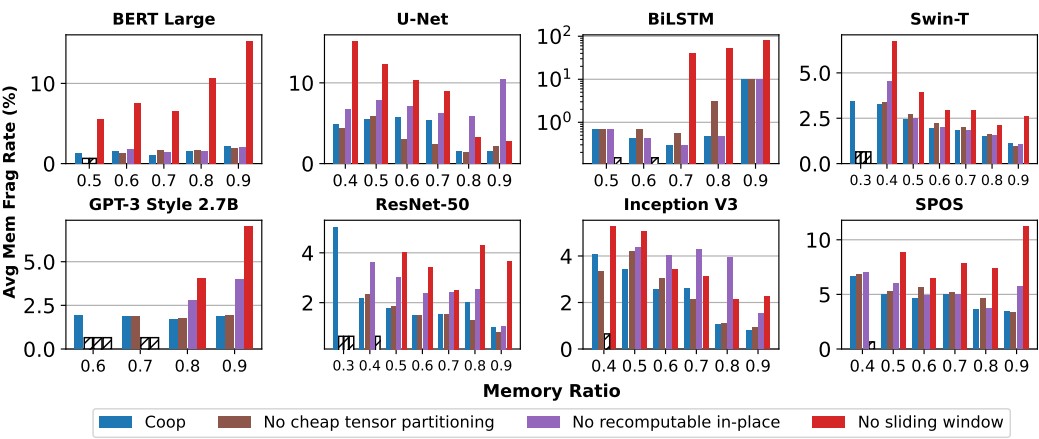

Figure 3: Comparison of the averaged memory fragmentation rate evaluated on Coop when one of the three modules is removed. The bars with slashes represent the OOM error.

# C   Cutoff Memory Budget

Figure 4 illustrates the cutoff memory budget, below which at least one tensor is evicted during training. With Coop, the cutoff memory budget is lower than DTR and DTE on all eight DNNs. This proves that Coop utilizes memory more efficiently. The recomputable in-place module in Coop promises that the unevictable tensors (*e.g.*, the parameters), which are allocated at the beginning of the training, are always kept at the side of the memory pool. Thus, the memory pool will not be

divided into pieces when the unevictable tensors are mutated. Another reason is that Coop also avoids redundant evictions when copying the tensors in in-place operations.

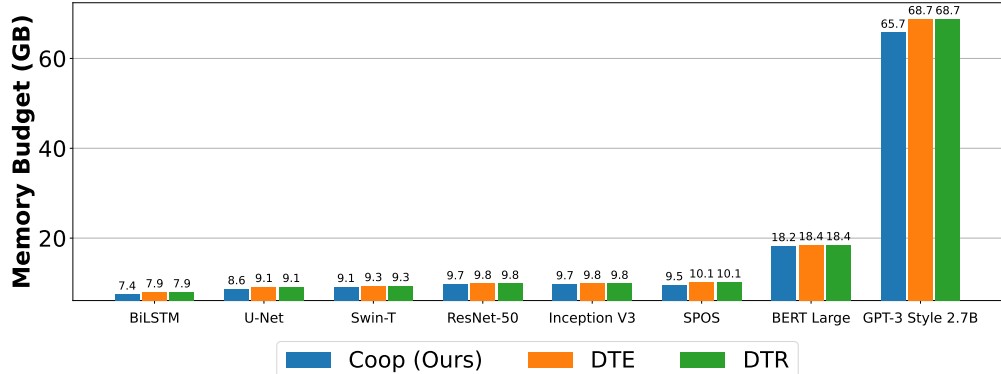

Figure 4: The cutoff memory budgets achieved by Coop, DTE, and DTR. At least one tensor is evicted during training when the threshold of memory is below the cutoff memory budget. For a given DNN, the lower the cutoff memory budget is, the more efficient GPU is utilized.

# References

[1] Zhongzhe Hu, Junmin Xiao, Zheye Deng, Mingyi Li, Kewei Zhang, Xiaoyang Zhang, Ke Meng, Ninghui Sun, and Guangming Tan. Megtaichi: dynamic tensor-based memory management optimization for dnn training. In *Proceedings of the 36th ACM International Conference on Supercomputing*, pages 1–13, 2022.