# OpenReview forum: "Coop: Memory is not a Commodity"
_NeurIPS.cc/2023/Conference — NeurIPS 2023 spotlight_

### Official Review · Reviewer_UBH8 · 2023-06-26

**Soundness:** 3 good
**Presentation:** 3 good
**Contribution:** 3 good
**Rating:** 7
**Confidence:** 5

**Summary:**

The authors propose to consider memory fragmentation when using tensor rematerialization on dynamic computation graphs. With three new memory management methods, sliding window algorithm, cheap tensor partitioning, and recomputable in-place, the authors enhance the checkpointing method with lower computation overhead and memory consumption.


**Strengths:**

* The paper is well-organized and easy to follow.
* The perspective is new. Considering memory fragmentation can enhance the gradient checkpointing.
* The results are great across the eight neural networks.
* The implementation is available in the supplementary material.


**Weaknesses:**

**Major issues**
1. Table 1. The cost density depends on the hardware accelerators. It is better to introduce the hardware specifications. The authors can also mention the arithmetic intensity, which is independent of the hardware. Also, in a single neural network, different operators belonging to the same category may have different cost densities (e.g., different kernel sizes in convolution layers).
2. In cheap tensor partitioning, the authors allocate tensors from the leftmost and rightmost of the memory pool. Is it possible to have more ports for the memory pool? What if there are 3 levels of cost densities in the computation graph? In general computation graphs, the cost density or the arithmetic intensity of operators has a continuous distribution. Is it a good idea to classify them into two categories, “cheap” and “expensive” tensors?
3. The authors miss one of the most important evaluation metrics, the end-to-end training time under different memory budgets, which is more critical than the computation overhead.
4. The authors mention that “static graph methods are beyond the scope of this paper.” I am aware that symbolic and eager executions have distinct features. However, I have several concerns about that.
    * Among the eight computation graphs in the experiments, six have static structures, while only two have dynamic ones. It is better to discuss more on the dynamic structures.
    * The proposed method leverages the global information that several tensors are unevictable. If the computation graph is fully dynamic, we have no idea which tensors will be used in the future.

**Minor issues**
1. Line 180. Please explain the N. I assume that it is the number of tensors.
2. A period after “Compute Overhead and Memory Budget” in Line 287, “Search Latency” in Line 303, and “Memory Fragmentation” in Line 315.
3. The caption of Figure 5. y-axes -> y-axis.
4. Results on Search Latency. The authors may list other statistics, e.g., min/max value, standard deviation to demonstrate, to demonstrate the small variations of the proposed method.
5. The authors mainly use the term “memory allocation” in the paper. It is better to replace it with “memory management” since there are other memory operations, such as eviction.


**Questions:**

Please see the Weaknesses, especially the major issues.

**Limitations:**

The authors do not discuss limitations. I think the proposed COOP has the inherent limitations of the checkpoint method on the dynamic computation graphs.

---

> ### Author Rebuttal · Authors · 2023-08-06
>
> We thank the reviewer’s appreciation for our work, and we address the questions below. We use W and L to denote major weakness and limitation correspondingly.
>
> > **W1:** It is better to introduce the hardware specifications about cost density. The authors can also mention the arithmetic intensity. Different operators belonging to the same category may have different cost densities.
>
>    We agree that the cost density depends on the hardware accelerators. The cost density in Table 1 is measured on NVIDIA GeForce RTX 2080 Ti. We will mention it in the revised manuscript.
>
>    Arithmetic intensity is the ratio of FLOPs to the summation of input and output sizes. In the revised manuscript, we will define a similar concept, FLOPs density (the ratio of FLOPs to output size), to match the heuristic of recomputation, and will show FLOPs densities and cost densities of various operators (e.g., convolutions with different kernel sizes). Some preliminary results are shown in Table 1 of the attached pdf.
>
> > **W2.1:** In general computation graphs, the cost density or the arithmetic intensity of operators has a continuous distribution. Is it a good idea to classify them into two categories, “cheap” and “expensive” tensors?
>
>    We agree with the reviewer that cost density continuously changes. We divided the operations into two categories since we observed an obvious jump in the cost densities of the commonly used operations, as shown in Figure 1 of the attached pdf.
>
>    Similar ideas of dividing the operations into two categories have been evidenced in Selective Activation Recomputation (SAR). SAR found that tensors generated by some operations (e.g., softmax and dropout) occupys 70% of the memory in GPT-3 while these operations only contribute to 2.7% FLOPs.
>
> > **W2.2:** Is it possible to have more ports for the memory pool?
>
>    It is possible to have more than two ports for the memory pool, where the operations are divided into two categories given the reasons in W2.1.
>
>    Figure 2(a)(1) shows an example of a 4-port memory pool with two 2-port sub-pools. The tensors are allocated to the two sub-pools alternatively. If there is no space left in one sub-pool, the tensors will be allocated to the other one. Figure 2(a)(2) shows an 8-port memory pool. Thus, we can construct memory pools with 16-port, 32-port, etc., in the same way.
>
>    If the ports are odd (2k+1), the memory pool is most likely to be consisted of k 2-port sub-pools and one 1-port sub-pool. This leads to insufficient usage of the memory as shown in Figure 2(a)(3).
>
>    The N-port memory pools can reduce the evictions of tensors that are continuous in the computational graph (thus reducing the heuristic) but will increase the memory fragmentation. We investigated this approach at the beginning of this work but did not apply it, since there is no significant improvement in the overall performance. Some experimental results of using 3-port and 4-port memory pools are shown in Figure 2(b). We will add these discussions to the revised manuscript.
>
> > **W3:** the end-to-end training time and the compute overhead.
>
>    We followed Checkmate and DTR to use the metric of compute overhead, which can be regarded as the normalized training time (the end-to-end training time with rematerialization divided by the end-to-end training time without rematerialization).
>
> > **W4.1:** It is better to discuss more on the dynamic structures.
>
>    We agree that online methods such as DTR and Coop are useful to optimize NNs with dynamic structures since offline optimization methods are not applicable to these dynamic NNs. We also studied multiple static NNs since (1) there are more static NNs than dynamic NNs, and (2) online methods also show advantages over offline methods when used to optimize static NNs. As in Reviewer BV8L’s comment, one popular offline method, Checkmate, is limited to running on small-scale networks. Our experiments for the Question 2 of Reviewer BV8L also show that Checkmate (with the most advanced commercial solver, gurobi) fails to find the best solution of ResNet-50 within a 14-hour time limit. This explains why PyTorch and TensorFlow only provide manual recomputation instead of the theoretically optimal method such as Checkmate. Given the small search latency of online methods such as DTR and Coop, we expect that applying online methods to optimize trainings of both dynamic or static NNs will be beneficial. We will add these discussions in the revised manuscript.
>
> > **W4.2:** Coop leverages the global information that several tensors are unevictable.
>
>    The unevictable tensors in Coop are the parameters and buffers in DNN. These tensors are available from the initialization of the network (e.g., the constructor of nn.Module in PyTorch and tf.keras.Model in TensorFlow 2.0). Therefore, Coop only uses information available for optimization.
>
> > **L1:** Coop has the inherent limitations of the checkpoint method on the dynamic computation graphs
>
>    We agree with the reviewer that Coop has some inherent limitations of online methods. We will add the discussions (as in the response to Reviewer 2’s L1) to the revised manuscript.
>
> **Minor issues**
>
> > **1.** The meaning of N at Line 180.
>
> The meaning of N is explained at Line 157. We will explain it at Line 180 in the revised manuscript to make it clearer.
>
> > **2.** List min/max value, standard deviation of search latency.
>
> Thanks for your valuable advice. We calculated these statistics (shown in the following table) and found that they are very useful to demonstrate the small variations of Coop.
>
> |   | Coop | DTE | DTR |
> | ------------- | ------------- | ---- | ---- |
> | Min  | 0.24  | 0.20 | 0.24 |
> | Max  | 18.6  | 13975.0 | 28019.0 |
> | Std  | 2.81  | 2349.1 | 4863.5 |
>
> > **3.** Missing periods, "y-axes" -> "y-axis" and "memory allocation" -> "memory management"
>
> We will update the manuscript accordingly.

---

> > ### Comment · Reviewer_UBH8 · 2023-08-11
> > **Thank you for the response**
> >
> > Thanks to the authors for their response, especially the attached tables and figures. Most of my concerns have been addressed. I have only one question regarding W2.1.
> >
> > > We divided the operations into two categories since we observed an obvious jump in the cost densities of the commonly used operations, as shown in Figure 1 of the attached pdf.
> >
> > > Coop does not make any assumptions about the structures of the neural network and can be universally implemented in any deep learning framework.
> >
> > The first statement is from the authors' response, and the second one is the last sentence of Section 5. I am aware that the field of MLSys focuses on both (1) general optimization without assumptions on model and hardware, and (2) oriented optimizations against a workload. The authors may define their contributions with a consistent claim.
> >
> > For most of the operators in NN, they can be simply classified into two categories by their complexity: linear or less (e.g., element-wise ops), larger than linear (matmul). I think this assumption makes sense and is widely adopted.
> >
> > Thank you for your great work.

---

> > > ### Author Response · Authors · 2023-08-11
> > > **Thanks for the constructive feedback**
> > >
> > > Thanks for the constructive feedback. We fully agree with the reviewer that most operators in neural networks can be classified into two categories by their complexities, i.e., linear/sub-linear and super-linear. This provides a theoretical basis for our experimental results (Figure 1 in the attached pdf).
> > >
> > > Therefore, Coop works under an implicit assumption that super-linear complexity translates to high cost density. Even though this assumption applies to most known neural networks, it is not theoretically guaranteed because of the constant terms. We will add the discussions about this assumption to Section 3.4 and update the statement that "Coop does not make any assumptions about the structures of the neural network" in Section 5 accordingly.

---

> > > > ### Comment · Reviewer_UBH8 · 2023-08-12
> > > > **Thanks for your great work**
> > > >
> > > > Thanks for your quick response. Given that the paper is solid and the implementations are submitted, I would like to raise my rating from 6 to 7.

---

### Official Review · Reviewer_ks3C · 2023-06-26

**Soundness:** 3 good
**Presentation:** 3 good
**Contribution:** 2 fair
**Rating:** 7
**Confidence:** 3

**Summary:**

Tensor materialization trades the memory with recomputation. Prior tensor materialization methods do not consider the memory fragmentation problem of the memory system used in deep learning frameworks, which makes them evict unnecesary tensors. The authors of this paper proposed a memory-system-aware rematerialization method called Coop to reduce the memory fragmentation. Experiments show that Coop can achieve up to 2x memory saving comparied with prior works.

**Strengths:**

1. The authors proposed a new method based on sliding window algorithm (sec 3.3) to alliviate the fragmentation problem of rematerialization.
2. The experiments show that this method is effective and can greatly reduce the compute overhead for specific memory ratio compared with prior works.



**Weaknesses:**

1. The requirement of underlying memory allocator might limit the applicability of the proposed method

This paper assumed the memory allocator used by the deep learning systems (discussed in section 2.1). The underlying memory allocator must be able to **merge** the freed chuncks if they are contiguous. There are other kinds of memory allocators (e.g., record the mapping from chunck size to a list of free chuncks with the specific chunk size) that do not have this feature, thus the proposed method can not be (directly) used for deep learning systems with this kind of memory allocators.

2. More discussion on the effectiveness of the the page-table-based memory system is needed

Similar to CPU memory system, the GPU memory system also employed the page table to manage its memory. Thus, we can free the discontiguous memory chunks and allocate a new one with the sum of the sizes of the freed chunks. From the virtual memory's view, the allocated memory is contiguous. Thus, there is no fragmentation problem discussed in this paper, and the prior works can be directly used.   The good news is that, since CUDA 11.2, we can directly use the memory pool [1] implemented in cuda runtime/driver to enjoy this feature. Thus, I am interested in whether the prior works have the fragmentation problem if they use the memory pool implemented in cuda?

We can use the following program to validate that the page-table based GPU memory system. In the program, we allocate 3 chuncks of 7 GB called p1, p2, and p3. Then, we free p1 and p3 (they are not contiguous). At last, we allocate a chunk of memory with 14 GB called p4 successfully.
```python
#include <stdio.h>
#include <cuda.h>
#include <unistd.h>

#define CHECK(e) {auto s = e; if (s != cudaSuccess) printf("CUDA error: %s", cudaGetErrorString(s));}
#define GB(x) ((x) * 1024ull * 1024 * 1024)

int main() {
    void *p1, *p2, *p3, *p4;
    CHECK(cudaMalloc(&p1, GB(7)));
    CHECK(cudaMalloc(&p2, GB(7)));
    CHECK(cudaMalloc(&p3, GB(7)));
    printf("first allocation done\n");
    printf("p1=%p\n", p1);
    printf("p2=%p\n", p2);
    printf("p3=%p\n", p3);

    // sleep 5 seconds, we can check the memory usage in `nvidia-smi` during this time
    sleep(5);

    printf("free p1 and p3, allocate p4\n");
    CHECK(cudaFree(p1));
    CHECK(cudaFree(p3));
    CHECK(cudaMalloc(&p4, GB(14)));
    printf("p4=%p\n", p4);

    printf("successfully allocated p4\n");
    CHECK(cudaFree(p2));
    CHECK(cudaFree(p4));
}
```
I can run above program on a RTX 3090 (24GB memory).
```
first allocation done
p1=0x7f4a68000000
p2=0x7f48a8000000
p3=0x7f46e8000000
free p1 and p3, allocate p4
p4=0x7f4528000000
successfully allocated p4
```

[1] https://docs.nvidia.com/cuda/cuda-runtime-api/group__CUDART__MEMORY__POOLS.html

**Questions:**

1. Whether the prior works have the fragmentation problem if they use the memory pool implemented in cuda? (See weaknesses part)


**Limitations:**

The proposed method relied on that the underlying memory allocator is able to merge contiguous chunks.

---

> ### Author Rebuttal · Authors · 2023-08-06
>
> We thank the reviewer’s appreciation for our work. Please see below our responses to your comments. We use Q, W, L to denote question, weakness, and limitation correspondingly.
>
> > **W1&L1:** The underlying memory allocator must be able to merge the freed chunks if they are contiguous.
>
> The reviewer's understanding is correct that Coop requires the memory allocator to be able to merge the freed chunk if they are contiguous. All DL frameworks have implemented 'merge', e.g., `try_merge_blocks` method in PyTorch's caching allocator and `Merge` method in TensorFlow's BFC allocator. Although 'merge' can only be used to combine the sub-block in the same block, it is common to apply for a single block in the size of the memory budget when rematerialization is enabled (so every sub-block is from the same block). Therefore, we suppose this limitation will not influence the wide application of Coop in most DL frameworks. We will add these discussions in the revised manuscript.
>
> > **W2&Q1:** More discussion on the page-table-based memory system is needed.
>
>    We are deeply appreciative of the reviewer's insightful comments. We modified the attached codes to use the memory pool implemented in CUDA:
>
>    ```cpp
>    #include <stdio.h>
>    #include <cuda_runtime_api.h>
>    #include <cuda.h>
>    #include <unistd.h>
>    #include <cassert>
>
>    #define CHECK(e) {auto s = e; if (s != cudaSuccess) printf("CUDA error: %s", cudaGetErrorString(s));}
>    #define GB(x) ((x) * 1024ull * 1024 * 1024)
>
>    int main() {
>        {
>            // assert current device supported memory pool
>            int value = 0;
>            CHECK(cudaDeviceGetAttribute(&value, cudaDevAttrMemoryPoolsSupported, 0));
>            assert(value == 1);
>        }
>        void *p1, *p2, *p3, *p4;
>        cudaStream_t stream;
>        CHECK(cudaStreamCreate(&stream));
>
>        CHECK(cudaMallocAsync(&p1, GB(2), stream));
>        CHECK(cudaMallocAsync(&p2, GB(2), stream));
>        CHECK(cudaMallocAsync(&p3, GB(2), stream));
>        printf("first allocation done\n");
>        printf("p1=%p\n", p1);
>        printf("p2=%p\n", p2);
>        printf("p3=%p\n", p3);
>
>        printf("free p1 and p3\n");
>        CHECK(cudaFreeAsync(p1, stream));
>        CHECK(cudaFreeAsync(p3, stream));
>        // Sleep 1 seconds to imitate the time between free and alloc.
>        // We can sleep arbitrary time here and the result is the same.
>        sleep(1);
>        printf("allocate p4\n");
>        CHECK(cudaMallocAsync(&p4, GB(2.1), stream));
>        printf("p4=%p\n", p4);
>
>        if (p4 != nullptr) {
>          printf("successfully allocated p4\n");
>          CHECK(cudaFreeAsync(p4, stream));
>        } else {
>          printf("failed to allocate p4\n");
>        }
>        CHECK(cudaFreeAsync(p2, stream));
>        CHECK(cudaStreamDestroy(stream));
>    }
>    ```
>
>    We ran the program on an NVIDIA GeForce RTX 2080 with 8GB memory and the chunk p4 **cannot** be successfully allocated. This indicates that the memory pool provided by CUDA did not fully utilize page table to optimize the memory usage. We further do three more experiments:
>
>    * **Experiment 1:** Replace `sleep(1)` with `cudaStreamSynchronize(stream)`. According to the CUDA documentation, `cudaStreamSynchronize` returns all free memory blocks in the CUDA memory pool to OS.
>
>       **Result:** The allocation of `p4` succeeds.
>    * **Experiment 2:** Free `p1` and `p2` (or `p2` and `p3`) instead of `p1` and `p3`, so that the freed memory is contiguous in the virtual memory space.
>
>       **Result:** The allocation of `p4` succeeds.
>    * **Experiment 3:** Based on Experiment 2, change the size of `p4` from 2.1GB to 4GB and 4.1GB, so that the desired memory size is exactly equal to or slightly larger than the freed memory size.
>
>       **Result:** The allocation of `p4` succeeds and fails, respectively.
>
>    These three experiments suggest that the memory pool provided by CUDA also caches the freed memory block, just like what the normal allocator of deep learning frameworks does. Based on these experimental results, we believe the prior works have the fragmentation problem even if they use the memory pool implemented in CUDA.
>
>    However, we fully agree that optimizing memory allocation by combining page table to reduce memory fragmentation is a promising direction. Studies have proven that memory fragmentation in CPU can be reduced by using similar ideas [1,2]. We believe we can apply the same idea to optimizing the general memory allocation (not only recomputation) of deep learning systems if we can manipulate the underlying operations of the GPU driver (rather than treating it as a proprietary NVIDIA-controlled black box). We will add these discussions to the revised manuscript.
>
>    [1] Maas M, Andersen D G, Isard M, et al. Learning-based memory allocation for C++ server workloads[C]//Proceedings of the Twenty-Fifth International Conference on Architectural Support for Programming Languages and Operating Systems. 2020: 541-556.
>
>    [2] Park C H, Cha S, Kim B, et al. Perforated page: Supporting fragmented memory allocation for large pages[C]//2020 ACM/IEEE 47th Annual International Symposium on Computer Architecture (ISCA). IEEE, 2020: 913-925.

---

> > ### Comment · Reviewer_ks3C · 2023-08-10
> >
> > Thanks the informative experiments on the memory pool provided by NVIDIA runtime/driver.
> >
> > I have no other questions. Good job!

---

### Official Review · Reviewer_bdP2 · 2023-07-03

**Soundness:** 3 good
**Presentation:** 3 good
**Contribution:** 2 fair
**Rating:** 6
**Confidence:** 2

**Summary:**

This paper proposes an optimization framework called Coop to solve the severe memory fragmentation, which is overlooked by prior tensor rematerialization works. Coop designs a sliding window algorithm to determine evicted tensor, guaranteeing the freed memory is contiguous and available for a new tensor.  Further, Coop adopts a cheap tensor partitioning method to rearrange the tensor in the memory layout based on the cost density, and a memory reuse mechanism, namely recomputable in-place, for the in-place operations. There two ideas are combined to reduce additional tensor rematerializaiton cost.

**Strengths:**

1. This paper provides a framework to solve the bottleneck of the high memory fragment rate in the tensor rematerialization scheme. The proposed sliding algorithm, cheap tensor partition mechanism, and recomputable in-place method reduce the rematerialization cost and improve memory utilization.
2. The problem formulation and writing make the paper is easy to understand.

**Weaknesses:**

1. It would be better to give an algorithm to describe the framework comprehensively.


**Questions:**

1. From the figures in the evaluation part, the proposed Coop is not always optimal in search latency. Please analyze the underlying reasons.
2. Is the additional sliding window search algorithm needed after recomputable in-place and cheap tensor partitioning in Figure 1? If not, how can the evicted tensors be determined with minimum cost? It would be better to provide an algorithm description.

**Limitations:**

1. Is there any trade-off for implementing the framework Coop?

---

> ### Author Rebuttal · Authors · 2023-08-06
>
> We thank the reviewer’s appreciation for our work. Please see below our responses to your comments. We use Q, W, L to denote question, weakness, and limitation correspondingly.
>
> > **W1:** It would be better to give an algorithm to describe the framework comprehensively.
>
>    We greatly appreciate your valuable comments. The pseudo-code algorithm for Coop is displayed below, with the core logic within the `allocate` function. The implementations of methods such as `free_and_merge_block` are omitted since they are straightforward and not directly related to the core logic. The pseudo-code will be added to Section 3.2 of the revised manuscript.
>
> ```python
> def evict(tensor):
>     # Release memory block and merge if possible
>     free_and_merge_block(tensor.addr)
>     tensor.addr = None
>
> def rematerialize(tensor):
>     if tensor.addr is not None:
>         return
>     for x in tensor.producer_op.inputs:
>         rematerialize(x)
>     run(tensor.producer_op)
>
> def run(op):
>     output_size = infer_size(op)
>     output = allocate(output_size, op)
>     # ...
>     if is_inplace_mutation(op):
>         op.inputs[0].addr = None
>
> def allocate(size, producer_op):
>     if is_inplace_mutation(producer_op):
>         # Apply recomputable in-place
>         # For simplicity, we assume in-place mutation operations have only one input tensor.
>         # The addr of the input tensor will be set to None at the end of `run` method.
>         addr = producer_op.inputs[0].addr
>     else:
>         block = find_free_block_large_than(size)
>         if block is None:
>             # Apply sliding window algorithm
>             evict(sliding_window_search(size))
>             block = find_free_block_large_than(size)
>         # Apply cheap tensor partitioning
>         if is_expensive(op_type):
>             addr = block.left_addr
>         else:
>             addr = block.right_addr - size
>     return Tensor(addr, size, producer_op)
> ```
>
> > **Q1:** Why is Coop not always optimal in search latency?
>
>    We provided an explanation in Line 310. Given the structure of ResNet-50, evicting a single resident tensor might be sufficient to allocate a new one. In this case, both DTR and DTE could find the best tensors to evict after one iteration. Therefore, the search latencies depend on the engineering implementation rather than the strategies themselves.
>
> > **Q2:** Is the additional sliding window search algorithm needed after recomputable in-place and cheap tensor partitioning in Figure 1? If not, how can the evicted tensors be determined with minimum cost?
>
>    The reviewer's understanding is correct that the memory layout is first optimized during tensor allocation by using recomputable in-place and cheap tensor partitioning. Sliding window algorithm is used to find the best tensors to evict given the current memory layout. The three modules are flexibly used during the whole training process. The sequence of the three modules in Figure 1 is not the sequence of using them in time. We will modify the caption of Figure 1 and add the pseudo-code in the answer of W1 to the revised manuscript to make the whole pipeline clearer.
>
> > **L1:** Is there any trade-off for implementing the framework Coop?
>
>    Coop comes with its own memory pool, so it cannot be simultaneously used with CUDA's built-in memory pool (stream-ordered memory allocator). The advantage of using the stream-ordered memory allocator is that multiple programs that use the stream-ordered memory allocator can share the same memory pool. However, all existing deep learning frameworks use their own memory pools instead of CUDA's built-in memory pool in default to achieve better efficiency and flexibility.
>
>    Additionally, Coop is an online method so it inherits the pros and cons of online methods. It can provide an efficient solution to finding the best tensors to evict within negligible time. However, the solutions may not match the optimal solutions as solved by the offline methods such as Checkmate, even though these offline methods usually require additional solvers and take several hours or multiple days.
>
>    We will add these discussions about limitations to the revised manuscript.

---

> > ### Comment · Reviewer_bdP2 · 2023-08-20
> >
> > The comments addressed my concerns. Thank you very much!

---

### Official Review · Reviewer_BV8L · 2023-07-07

**Soundness:** 2 fair
**Presentation:** 2 fair
**Contribution:** 3 good
**Rating:** 6
**Confidence:** 3

**Summary:**

This paper considers the rematerialization problem for DNN training and studies it from the perspective of memory fragmentation.

**Strengths:**

Please see the "Questions" section.

**Weaknesses:**

Please see the "Questions" section.

**Questions:**

- I think this paper is interesting in the sense that it raises and studies a problem that could affect the performance of other rematerialization algorithms in the literature. The problem of memory fragmentation is not taken into account in most of the DNN memory optimization papers in the literature.

- The comparisons presented in the paper seem to be limited to heuristic based methods only. I think a comparison against the checkmate method ([11]) would make the results more interesting. The checkmate method is known to return the optimal solution since it's an exact method. However, it's also known that it doesn't scale to large-scale graphs. Perhaps numerical experiments for some small scale graphs could be still valuable since checkmate as a baseline would represent the optimal under the assumption that memory fragmentation is not an issue. This, in my opinion, would make the contributions of this paper clearer.

- I haven't read the work of [21]. Given what the authors discuss about the DTE method, is this statement in line 64 true: "We argued for the first time that existing tensor rematerialization methods overlook the memory system during optimization and wrongly assume that the memory in DL systems is a fungible commodity"? Another statement similar to this is "To the best of our knowledge, Coop is the only tensor rematerialization scheme that fully bypasses the incorrect assumption of DL memory system."

Minor
- It is not immediately clear what is meant by "search latency" the first time it is mentioned in the text. It is explained later in the text, perhaps that explanation could be moved up to where it's mentioned first in the text.
- This sentence in line 70 is hard to follow: "The properties of memory allocators in deep learning frameworks are considered to reduce the heuristic ..."

**Limitations:**

Please see the "Questions" section.

---

> ### Author Rebuttal · Authors · 2023-08-06
>
> We thank the reviewer’s appreciation for our work. Please see below our responses to your comments.
>
> > **Q2:** Comparison between Coop and Checkmate
>
> We agree that a comparison with Checkmate could further demonstrate Coop's contributions. Checkmate's publicly available code includes two git branches, namely mlsys20_artifact and master.
>
> The mlsys20_artifact branch is designated for replicating experiments from their paper. As the reviewer mentioned, these experiments do not account for memory fragmentation. What's more, Checkmate in this branch cannot generate executable networks (as evident in the tests/test_execution.py file within the mlsys20_artifact branch). Consequently, it cannot be used to investigate the impact of memory fragmentation.
>
> The master branch is capable of generating executable TensorFlow2 Keras computational graphs. Running these actual Keras graphs should help us comprehend memory fragmentation's effects on Checkmate. However, in the case of ResNet50, we ran Checkmate for 14 hours without obtaining any results within a single designated memory budget, despite leveraging the advanced MILP solver, Gurobi. Furthermore, attempts to optimize U-Net using Checkmate yielded nearly identical networks regardless of the budget value specified, rendering the collected data nonsensical.
>
> We will continue to make ongoing efforts, aiming to include a Coop and Checkmate comparison in the revised manuscript's appendix. Additionally, reviewers can refer to the comparison experiments between DTR and Checkmate in [1]. These experiments also do not account for memory fragmentation and do not generate executable networks. They reveal that DTR and Checkmate exhibit comparable performance across three networks (VGG-16, MobileNet, U-Net). In our experiments considering memory fragmentation, Coop outperforms DTR significantly (Figure 4), and DTR displays pronounced memory fragmentation issues (Figure 5). We believe this can serve as a supplementary rough reference.
>
> > **Q3:** Is Coop the only tensor rematerialization scheme that fully bypasses the incorrect assumption of DL memory system?
>
> The heuristic in DTE encourages evictions of tensors that are adjacent to free memory blocks. However, DTE is still a greedy algorithm that runs in a loop to find the best tensor to evict until the next tensor can be successfully allocated. This brings in the redundant and discontinuous evictions, as a limitation of assuming the memory in DL systems is a fungible conmmodity. In comparison, Coop optimizes 'tensor allocation' and uses the sliding algorithm to produce a continuous block. Therefore, we claimed that Coop is the first to fully bypass this incorrect assumption.
>
> **Minor points:**
>
> > **1.** It is not immediately clear what is meant by "search latency" the first time it is mentioned in the text. It is explained later in the text, perhaps that explanation could be moved up to where it's mentioned first in the text.
>
> We will define search latency at the position of its first occurrence.
>
> > **2.** This sentence in line 70 is hard to follow: "The properties of memory allocators in deep learning frameworks are considered to reduce the heuristic ..."
>
> We will rephrase this sentence as: The heuristic of tensor rematerialization is reduced by taking into account the properties of memory allocators, and the memory allocators are improved by considering the efficiency of different operations in tensor rematerialization.
>
> [1] Kirisame M, Lyubomirsky S, Haan A, et al. Dynamic tensor rematerialization[J]. arXiv preprint arXiv:2006.09616, 2020.

---

> > ### Comment · Reviewer_BV8L · 2023-08-15
> >
> > Thanks for the responses to my questions. I still believe that the insights of this paper on the memory fragmentation aspect of rematerialization are important and therefore I maintain a positive opinion about the work.

---

### Author Rebuttal · Authors · 2023-08-10

We thank the reviewers' appreciation and valuable advice for our work. Some new figures and tables are in the attached pdf file.

---

### Decision · Program_Chairs · 2023-09-21

**Decision:**

Accept (spotlight)

**Comment:**

The paper introduces "Coop", an optimization framework aimed at addressing the memory fragmentation challenge in tensor rematerialization for Deep Neural Network (DNN) training. This is achieved through a combination of sliding window algorithm, cheap tensor partitioning, and recomputable in-place mechanisms. The paper claims that Coop reduces memory costs and improves memory utilization.

**Strengths:**

- Addresses the overlooked issue of memory fragmentation in tensor rematerialization.
- Introduces a unique approach with the sliding window algorithm, tensor partitioning, and recomputable in-place method to improve memory management.
- Clear problem formulation and writing make the paper comprehensible.
- Empirical results demonstrate up to 2x memory savings compared to previous approaches.
- The manuscript is well-organized and clear, with supplementary implementation materials provided.

**Weaknesses:**

- Unclear terminologies at places, such as "search latency".
- The paper is often unclear in terms of hardware specifications, tensor partitioning, and end-to-end training time under different memory budgets.
- The paper does not delve deeply into the potential benefits and limitations of using the page-table-based GPU memory system.
- Other limitations need to be described more explicitly. For example: (a) it relies on a specific type of memory allocator that can merge contiguous chunks, potentially limiting its applicability to certain deep learning systems. (b) The focus seems more inclined towards static computation graphs. Any limitations around dynamic computation graphs need to be described.
- Limited comparisons in the paper. Notably, a comparison against the "checkmate" method is missing, which represents an optimal solution.

The paper offers valuable contributions and fresh perspectives on tensor rematerialization by addressing memory fragmentation. However, there are areas that require further clarification and explicit description of limitations. I strongly recommend the authors revise the paper to accommodate these weaknesses.